# Complications of Tumor Nephrectomy with and Without Tumor Thrombus in the Vena Cava, Recorded with the Clavien–Dindo Classification: A Matched-Pair Analysis

**DOI:** 10.3390/cancers16203523

**Published:** 2024-10-18

**Authors:** Ute Maria Frölich, Katharina Leucht, Marc-Oliver Grimm, Susan Foller

**Affiliations:** 1Department of Urology, Jena University Hospital, 07747 Jena, Germany; ute-froelich@web.de (U.M.F.); katharina.leucht@med.uni-jena.de (K.L.);; 2Comprehensive Cancer Center Central Germany (CCCG), 07743 Jena, Germany

**Keywords:** complications, Clavien–Dindo classification, nephrectomy, tumor thrombus, thrombectomy, renal cell carcinoma

## Abstract

**Simple Summary:**

This study looked at the complications that can occur following nephrectomies for kidney cancer with and without a so-called tumor thrombus, a tumor that has spread into the large vein called the inferior vena cava. So far, insufficient data are available on this topic. We developed a new way to classify the different types of complications, which, along with the established Clavien–Dindo classification, helped to better analyze these complications. We reviewed the data of 88 patients and found that the 44 patients with a tumor thrombus experienced significantly more and more severe complications than the 44 patients without a tumor thrombus. These results highlight the importance of having these complicated surgeries performed in specialized hospitals by experienced urologists and skilled staff. This practice could lead to earlier detection, better prevention, and more effective treatment of any complications arising after such surgeries. Consequently, the research findings could improve clinical practice and optimize treatment outcomes for patients undergoing nephrectomies.

**Abstract:**

Background/Objectives: Radical nephrectomy (RN) with inferior vena cava thrombectomy (IVCT) is indicated for the curative management of renal cell carcinoma (RCC) with tumor thrombus (TT). In the literature, any direct comparison of complications between RNs with or without IVCT is lacking. The objective of this study was to analyze and compare complications after RNs with or without IVCT. Methods: A retrospective evaluation of the complications recorded in RCC patients who underwent RN with (TT group, *n* = 44) or without (non-TT group, *n* = 44) IVCT between 2009 and 2021 was conducted. The non-TT group was identified via propensity-score matched-pair analysis. Postoperative complications up until discharge or postoperative day 30, whichever came first, were classified using the Clavien–Dindo classification (CDC). Complications were categorized into cardiovascular, pulmonary, bleeding, gastrointestinal, neurological/psychiatric, wound, urinary tract, dysglycemia, and other groups. Statistical analyses using descriptive statistics included the chi^2^ and Mann–Whitney U tests. Results: All CDC-grade postoperative complications were more frequent in the TT than in the non-TT group regarding the number of patients affected (93% vs. 73%), as well as per patient (median: 3 vs. 1; *p* < 0.001). Complications in CDC grade ≥ 3 were rare and comparable between groups. Cardiovascular, gastrointestinal, neurological/psychiatric, and bleeding complications occurred significantly more often in the TT group. However, its small study population and retrospective character limit this study. Conclusions: Significantly more patients undergoing an RN-IVCT experience more frequent postoperative complications than patients with an RN but without IVCT. Surgeons performing the procedures should be experienced, and hospital staff should be trained in the early recognition and treatment of complications.

## 1. Introduction

Renal cell carcinoma (RCC) is the ninth most common tumor worldwide [1]. At diagnosis, about one-third of RCCs are metastatic [2,3,4]. RCC is also associated with tumor extension into the blood vessels: 4% to 10% of patients present with a tumor thrombus (TT) extending into the renal vein or inferior vena cava (IVC) [5]. The Mayo classification is a common method for grading a TT by its extension [6,7]. If the TT extends into the IVC and remains untreated, the median life expectancy is 5 months, and 1-year disease-specific survival is 29% [8]. Despite its surgical complexity, radical nephrectomy with IVC thrombectomy (RN-IVCT) remains the standard treatment for an RCC with TT [9,10,11]. Preoperative considerations, such as using modern imaging techniques to evaluate the proximal extent, the volume, and the potential caval wall invasion of the TT, are essential. In some cases, extracorporeal circulation becomes necessary. After RN-IVCT, complications are significantly more frequent than after nephrectomy alone. In particular, transfusions, acute kidney injury, and cardiac and thromboembolic events are likely [12]. The hospital and surgeon volumes of such procedures significantly affect survival [13,14,15]. Thus, it is recommended to admit patients to high-volume centers that perform at least three procedures per year [13].

Complication reporting should be standardized to allow for inter-individual comparability [16]. Different classification systems are in use, each having advantages and disadvantages [17]. The Clavien–Dindo classification (CDC), a five-level gradation system that is dependent on the treatment applied, is broadly accepted [18] and is validated in urology by the Ad-Hoc EAU Guidelines Panel [19]. The latter lists important aspects that should not be missed when reporting complications.

We aimed to identify complications after RN-IVCT, to report them in a standardized manner using the CDC system, following the recommendations of the Ad-Hoc EAU Guidelines Panel, and to compare them in a matched-pair analysis with complications emerging after RN alone. This comparison between matched pairs of patients with and without TT allows for the determination of differences by controlling the effects of other characteristics.

## 2. Materials and Methods

### 2.1. Study Design and Data Collection

For this retrospective case-control study, we identified patients who underwent radical transperitoneal nephrectomy for RCC at Jena University Hospital, using procedure codes. Data were collected by chart review from an electronic documentation system (SAP SE & Co. KG, Walldorf, Germany, version 8000.1.8.1161) and paper archive files.

From the initial group, we excluded some individual cases, such as those without transperitoneal access or those with additional complex surgery, since these circumstances may have affected the outcome of surgery (Figure 1). As the smallest diameter of a tumor in the TT group was 3.8 cm, we excluded 10 cases in the control group with a smaller tumor diameter. For each of the 44 patients with TT in the IVC (TT group), 1 patient was propensity score-matched from the remaining 112 patients without TT (control group), creating a non-TT group (n = 44). All study data were recorded in accordance with the local ethics committee requirements.

### 2.2. Preoperative Data and Risk Factors

We assessed the patients’ health status using the ASA score (American Society of Anesthesiologists) [20]. We used the Charlson Comorbidity Index (CCI) to assign scores based on comorbidities and age [21,22]. To classify the spread of TT, the Mayo classification (Appendix A) was applied [6].

### 2.3. Surgical Procedure

In all cases, a transperitoneal approach was adopted. The RN-IVCTs were conducted by experienced urologists [23]. In contrast, radical nephrectomy for non-TT patients was part of the training program for less-experienced urologists. If the TT extended cranially beyond the diaphragm into the right atrium (T3c, Mayo level 4), the procedure was carried out in collaboration with cardiothoracic surgeons.

### 2.4. Postoperative Data and Definition of Standard Postoperative Course

We focused on early postoperative complications until discharge or until postoperative day 30, whichever came first. A standardized postoperative course was defined, with the following measures considered as standard: treatment with antiemetics (except with records of nausea or emesis), the application of analgesics, laxative measures (except with records of obstipation after at least 3 days post-surgery), treatment with catecholamines (≤ 24 h), non-invasive ventilation (without interruption after surgery, only until the first (ventilation) or second (non-invasive ventilation) postoperative day). Interpreting the CDC strictly would mean including electrolyte imbalances with the substitution of electrolytes as a complication, which affected more than two-thirds of our patients. In similar publications, electrolyte imbalances with the substitution of electrolytes were not listed as complications according to the CDC guidelines. To ensure comparability, we excluded these, too (Appendix A).

### 2.5. Grading and Reporting of Complications

We graded each complication according to the CDC system, which defines severity based on the medical intervention required [18]. We categorized the complications into cardiovascular, pulmonary, bleeding, urinary, gastrointestinal, neurological/psychiatric, dysglycemia, wound, and other issues, and assigned them CDC grades. We aimed at the uniform reporting of complications according to the Ad-Hoc EAU Guideline Panel recommendations (Appendix A) [19].

### 2.6. Statistical Analyses

We used SPSS^®^ Statistics (IBM^®^, version 27, Armonk, NY, USA) for the statistical analyses. Propensity score matching was applied to select cases from the control group as counterparts for the 44 cases in the TT group, creating a non-TT group (n = 44). To determine the balance and success of propensity score matching, the standardized difference for each potential matching parameter (renal side, tumor location, anticoagulation, ASA-score, tumor diameter, age, and BMI) was calculated, containing information about the mean and variance per group. The following values were below the cut-off of d = 0.1, confirming the balance of the two groups [24]: ASA score: d = 0.061; renal side (right/left): d < 0.001; tumor location (upper pole/non-upper pole): d = 0.046; anticoagulation (yes/no): d = 0.075. Accordingly, these four were used as matching parameters.

Descriptive statistics were used to obtain the frequencies and percentages for nominal variables. Medians, means, and ranges were obtained for the metric and ordinal variables. Variables of the TT group and non-TT group were evaluated with the Mann–Whitney U test, chi^2^ test, or Fisher’s exact test. Statistical significance was set at *p* < 0.05.

## 3. Results

### 3.1. Preoperative Patient and Tumor Characteristics

At Jena University Hospital, 206 patients underwent radical transperitoneal nephrectomy for RCC between January 2009 and May 2021. As displayed in the CONSORT diagram above (Figure 1), we considered 44 TT patients and 44 non-TT patients for this matched-pair analysis. At Jena University Hospital, a median of 3 (range: 1–10) RN-IVCTs are performed per year. This makes it a high-volume center [13] (Appendix A).

The median patient age was 68 years (range: 44–87) at surgery; 45% were female. The median ASA score was 3 (1–4), the median BMI was 27 kg/m^2^ (18–46), and the median CCI was 6 (2–16). Of the patients, 34% had metastases at the time of diagnosis. In 75%, the tumor was right-sided and was located at the upper pole in 42%. Detailed baseline characteristics are displayed in Table 1.

Baseline and tumor characteristics were well-balanced between the groups, except for type II diabetes mellitus (TT group: 39% vs. the non-TT group: 14%, *p* = 0.008), macrohematuria (36% vs. 11%, *p* = 0.006), lung metastases (41% vs. 20%, *p* = 0.037), positive lymph nodes (28% vs. 9.1%, *p* = 0.029), fat infiltration (57% vs. 30%, *p* = 0.010), and residual tumor (39% vs. 11%, *p* = 0.003; usually vessel margins in the TT group). Additionally, in the TT group, the median tumor diameter was significantly larger than in the non-TT group (9.0 cm (range: 3.8–17) vs. 7.0 cm (4.1–15), *p* = 0.002). According to the Mayo classification, the TT grade was level 1 in 20%, level 2 in 25%, level 3 in 39%, and level 4 in 16% of TT patients.

### 3.2. Intra- and Postoperative Data

In TT patients vs. non-TT patients, the median operation time was significantly longer (239 min vs. 141 min, *p* < 0.001), median blood loss was significantly higher (900 mL vs. 300 mL, *p* < 0.001), and more patients needed intraoperative blood transfusions (55% vs. 14%, *p* < 0.001), with a significantly higher median number of units per patient (*p* < 0.001, Table 2). Anticoagulation was applied in 11% of TT patients and 9.1% of non-TT patients (*p* = 1.000). Extracorporeal circulation was used in all four cases with TTs in the atrium.

The median postoperative hospital stay was comparable between groups (TT group: 11 nights vs. non-TT group: 10 nights, *p* = 0.207). In the TT group, significantly more TT patients than non-TT patients stayed at an intensive care unit (ICU) for at least 1 night (30% vs. 6.8%, *p* = 0.006) with a significantly longer median length of stay (*p* = 0.006). Significantly more TT patients than non-TT patients needed postoperative transfusions (50% vs. 23%, *p* = 0.008), with a significantly higher median number of units per patient (*p* = 0.012). Treatment with catecholamines was applied significantly more often in TT patients vs. non-TT patients (39% vs. 11%, *p* = 0.003) and for longer periods (*p* = 0.004). However, the duration of catecholamine administration was > 24 h in only 14% of TT patients and 6.8% of non-TT patients (*p* = 0.484). TT patients required significantly more (32% vs. 6.8%, *p* = 0.006) and longer (*p* = 0.003) postoperative ventilation than non-TT patients (Table 2).

### 3.3. Postoperative Complications

Each complication until discharge or postoperative day 30, whichever came first, was categorized according to its nature and assigned a CDC grade (Table 3). Of the patients, 93% (TT group) and 73% (non-TT group) had at least one complication (*p* = 0.021). In the TT vs. non-TT group, 6.8% vs. 27% had no complications, 6.8% vs. 18% had a maximum CDC grade 1, 75% vs. 45% had a maximum CDC grade 2, and 11% vs. 9.1% had a maximum CDC grade ≥ 3. No complication lasted longer than their hospital stay (no suffix ‘d’ according to CDC). No patient died during the observation period.

The most common complications were bleeding (TT group: 66% vs. non-TT group: 32%), cardiovascular (57% vs. 30%), gastrointestinal (50% vs. 23%), urinary tract (27% vs. 30%), and pulmonary (28% vs. 18%). Cardiovascular, bleeding, gastrointestinal, neurological/psychiatric, and “other” complications occurred significantly more often in the TT group, whereas wound complications occurred significantly more often in the non-TT group. Each patient had a median of 3 (range: 0–20, TT group) or 1 (0–16, non-TT group) postoperative complications in the observed period (Table 4).

Most complications were classified as CDC grade 1 due to the use of antiemetics (TT group vs. non-TT group: 39% vs. 20%), specific physiotherapy (30% vs. 16%), and diuretics (25% vs. 20%). Apart from additional wound treatment, which was applied in significantly more TT patients than non-TT patients (2.3% vs. 20%, *p* = 0.015), CDC grade 1 medications were applied without significant differences between groups. As for CDC grade 2, significantly more TT patients than non-TT patients required medication, other than for grade 1 medication (80% vs. 50%, *p* = 0.004) and blood products (52% vs. 23%, *p* = 0.004). For CDC grades 3 and 4, no differences between groups emerged (Table 4).

## 4. Discussion

RN-IVCT is the treatment of choice for the curative management of RCC with TT in the IVC [25]. According to the literature, RN-IVCT is an effective cancer control procedure with acceptably low mortality and morbidity rates in experienced hands [26]. Complication rates depend on the extent of tumor vascular invasion and the complexity of the IVC reconstruction [12,26]. Accordingly, in our matched-pair analysis, complications were significantly more frequent for RN-IVCT than for nephrectomy alone.

If patients present with a TT, it is recommended that they be admitted to high-volume centers [13,14,15]. A significant difference in median overall survival of 42, 53, and 60 months has been reported for low-, medium-, and high-volume centers, respectively (*p* = 0.009) [13]. The case volume also correlates with mortality in other major urological procedures, such as radical cystectomy for bladder cancer [27].

To optimize postoperative management and adjust treatment strategies, a careful analysis of the perioperative period of RN-IVCT cases is warranted. However, the literature on this topic is limited due to the lack of control groups [28] or the non-standardized reporting of complications [12]. Frequently, minor complications are not reported. Uniform reporting is a prerequisite for inter-individual comparisons. Therefore, we present a direct comparison (matched-pair analysis) between the complications of TT patients and non-TT patients, following the recommendations of the Ad Hoc EAU Guidelines Panel [19].

No deaths and few major complications of CDC grade ≥ 3 occurred in both TT and non-TT patients, indicating that a nephrectomy with or without IVCT is a safe procedure. However, there were significant differences between groups regarding patients without and with complications of CDC grade < 3 only. Of TT vs. non-TT patients, 6.8% vs. 18% had maximum CDC grade 1 complications and 75% vs. 45% had maximum CDC grade 2 complications. If we had summarized and reported these complications as minor, the differences between the groups would have been smaller, and important details would have been lost. This supports previous statements suggesting that CDC grading is preferable to solely reporting minor and major complications [19]. Looking more closely at grade 2-level complications, it can be seen that TT patients required both blood products and grade 2 medications significantly more often, information that would have been overlooked if the CDC grading system had not been used. In many cases, the grading of complications was relatively simple and precise. In others, however, the categorization was challenging. One example is the substitution of electrolytes, which was applicable in about two-thirds of our patients. We believe that it was often not associated with any complication but was associated with standard management practices after major surgery. Therefore, we have defined a standard course of nephrectomy with or without IVCT that excludes certain incidents, which, according to a strict interpretation of the CDC, would have counted as a complication.

Intraoperatively, larger tumor diameters and the preparation of the IVC were associated with greater surgical effort, longer operation time, higher blood loss, and more intraoperative transfusions (number of patients and units needed) in the TT group. Thrombectomy and adjacent organ removal were significant predictors for intraoperative complications according to multivariate analysis, and estimated blood loss predicted all grades of postoperative complications [15].

In the literature, the 30-day mortality for RN-IVCT is relatively low (1.5–10%), but early postoperative morbidity is considered high (15–78%) [26]. Similar to our cohort, a single-arm study (n = 61; Mayo level 2–4) reported that patients were predominantly affected by low CDC-grade complications: grade 1: 38%; grade 2: 38% [28]. Of these patients, 16% had complications of a grade ≥ 3b and 3.3% (n = 2) died. This appears slightly higher than in our TT group (11% grade ≥ 3; no deaths). Unfortunately, there was no information provided about the number of patients without any complications, and information was also lacking for the control group.

Hennus et al. [29] highlighted that patient-related factors are considered important in terms of surgical outcome and prognosis after nephrectomies. They confirmed the direct impact of major comorbidities on postoperative complications while, e.g., obesity and previous abdominal surgery were not identified as risk factors. In their cohort, 51% had preoperative comorbidities (CCI ≥ 1) and 36% exhibited CCI 1 or 2. The overall complication rate was 34%. A median CCI of 6 in our analysis reveals a potential selection bias for our university hospital cohort, including many highly morbid patients. This fact, as well as our stringent documentation of each deviation from the normal surgical course, might explain the relatively high overall rate of complications (83%) in our cohort.

We categorized each complication by its nature. This technique has been adopted by others before, but they used a categorization that differed from ours, probably due to the number and appearance of single complications, or due to individual preference [12,15,29,30,31]. The fact that we attributed electrolyte imbalances to the normal postoperative course improves comparability with previous publications, as electrolyte replacement was not listed as a CDC complication in those publications either [12,15,29,30,31]. However, electrolyte imbalances with the substitution of electrolytes appeared significantly more often in non-TT patients than in TT patients. This finding might be attributed to the imprecise documentation of mild complications such as electrolyte imbalances in TT patients when other, more serious complications occurred concurrently, which masked less serious complications.

Our study has limitations, primarily due to the small study population and its retrospective character. The important and possibly outcome-affecting baseline characteristics of tumor size and the presence of lung metastases were not balanced between groups. This was possible because both parameters were not included in the propensity score matching. To determine the balance and success of the propensity score-matching process, the standardized difference was calculated for each potential matching parameter. Tumor size was taken into account but did not emerge as a matching parameter. Metastases and their location were not included in the initial calculations. Another limitation is the non-reporting of later complications. This decision was made due to the high level of missing data after patients were discharged from hospital. We suspected a bias due to under-documentation, e.g., in the reports from rehabilitation centers. A prospective, multicenter study design, a larger study population, and a longer follow-up period would be desirable in future studies.

## 5. Conclusions

The present matched-pair analysis provides meticulous insights into the expected postoperative complications following RN-IVCT. Complications were documented stringently and in a highly standardized manner using the CDC. In addition to frequency and severity, our data enable the direct comparison of complications after RN with or without IVCT. Our analysis indicates that significantly more patients undergoing RN-IVCT experience more frequent postoperative complications than patients with RN without IVCT; therefore, they require more intensive monitoring and treatment. Cardiovascular, bleeding, gastrointestinal, neurological/psychiatric, and “other” complications occurred significantly more often in the TT group. Surgeons should be experienced in this type of surgery and hospital staff should be trained and routinized to recognize and treat potential complications early.

## Figures and Tables

**Figure 1 cancers-16-03523-f001:**
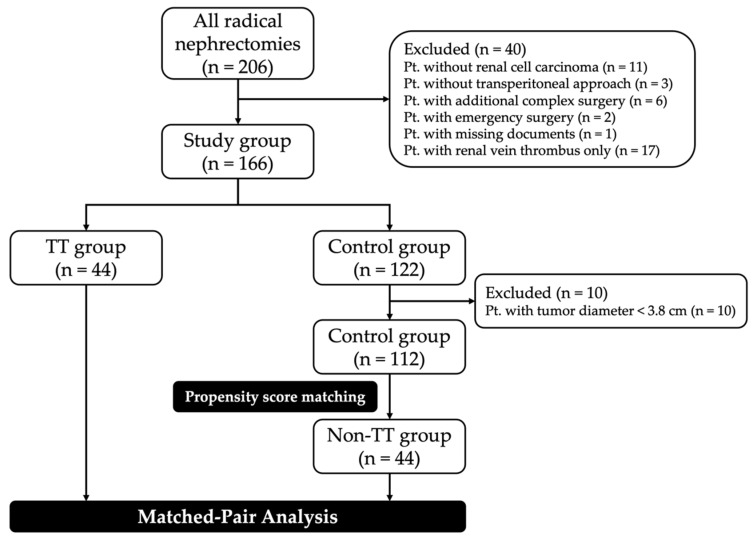
CONSORT diagram of the analysis. Pt—patient(s); TT—tumor thrombus.

**Table 1 cancers-16-03523-t001:** Preoperative patient characteristics and tumor characteristics.

Preoperative Patient Data	TT Group	Non-TT Group	Overall	*p*-Value
	(n = 44)	(n = 44)	(n = 88)	(TT vs. Non-TT)
Median age, years (range)	68 (44–84)	66 (45–87)	68 (44–87)	0.517
Female patients, n (%)	19 (43)	21 (48)	40 (45)	0.669
Median BMI, kg/m^2^, (range)	27 (18–40)	27 (19–46)	27 (18–46)	0.930
Median ASA-score (range)	3 (2–4)	3 (1–4)	3 (1–4)	0.663
Metastases (preoperative), n (%)	18 (41)	12 (27)	30 (34)	0.177
Bone metastases, n (%)	5 (11)	1 (2.3)	6 (6.8)	0.202
Lung metastases, n (%)	18 (41)	9 (20)	27 (31)	0.037
Liver metastases, n (%)	2 (4.5)	2 (4.5)	4 (4.5)	1.000
Soft tissue metastases, n (%)	2 (4.5)	0 (0)	2 (2.3)	0.494
Median Charlson comorbidity index (range)	7 (2–16)	5 (2–10)	6 (2–16)	0.087
Diabetes mellitus, type II, n (%)	17 (39)	6 (14)	23 (26)	0.008
Macrohematuria, n (%)	16 (36)	5 (11)	21 (24)	0.006
Tumor characteristics				
Median largest tumor diameter, cm (range)	9 (3.8–17)	7 (4.1–15)	8.4 (3.8–17)	0.002
Right-sided tumor, n (%)	33 (75)	33 (75)	66 (75)	1.000
Tumor location: upper pole of kidney, n (%)	19 (43)	18 (41)	37 (42)	0.829
Fat infiltration, n (%)	25 (57)	13 (30)	38 (43)	0.010
Mayo classification of TT				
Level 1, n (%)	9 (20)	n.a.	n.a.	n.a.
Level 2, n (%)	11 (25)	n.a.	n.a.	
Level 3, n (%)	17 (39)	n.a.	n.a.	
Level 4, n (%)	7 (16)	n.a.	n.a.	
TNM Classification				
T = primary tumor				
T1, n (%)	0 (0)	18 (41)	18 (20)	<0.001
T2, n (%)	0 (0)	12 (27)	12 (14)	
T3a, n (%)	0 (0)	12 (27)	12 (14)	
T3b, n (%)	37 (84)	0 (0)	37 (42)	
T3c, n (%)	4 (9.1)	0 (0)	4 (4.5)	
T4, n (%)	3 (6.8)	2 (4.5)	5 (5.7)	
N = lymph nodes *				
N0, n (%)	31 (72)	40 (91)	71 (82)	0.029
N1/2, n (%)	12 (28)	4 (9.1)	16 (18)	
M = metastases, n (%)	19 (43)	12 (27)	31 (35)	0.118
R = residual tumor				
R0, n (%)	27 (61)	39 (89)	66 (75)	0.003
R1/2, n (%)	17 (39)	5 (11)	22 (25)	
Tumor stage				
Tumor stage I, n (%)	0 (0)	16 (36)	16 (18)	<0.001
Tumor stage II, n (%)	0 (0)	7 (16)	7 (8)	
Tumor stage III, n (%)	24 (55)	9 (20)	33 (38)	
Tumor stage IV, n (%)	20 (45)	12 (27)	32 (36)	

BMI = body mass index; ASA = American Society of Anesthesiologists; TT = tumor thrombus; n.a. = not available; * Overall n = 87, TT group n = 43.

**Table 2 cancers-16-03523-t002:** Intra- and postoperative data.

	TT Group	Non-TT Group	Overall	*p*-Value
	(n = 44)	(n = 44)	(n = 88)	(TT vs. Non-TT)
Intraoperative data				
Median operation time, min (range)	239 (91–457)	141 (67–321)	179 (67–457)	<0.001
Anticoagulation, n (%)	5 (11)	4 (9.1)	9 (10)	1.000
Median blood loss, ml (range) ^1^	900 (100–5500)	300 (20–6000)	500 (20–6000)	<0.001
Intraoperative transfusions, n (%)	24 (55)	6 (14)	30 (34)	<0.001
Median intraoperative transfusion, units (range)	1 (0–9)	0 (0–7)	0 (0–9)	<0.001
Extracorporeal circulation, n (%) ^2^	4 (9.1)	0 (0)	4 (4.5)	0.116
Cavotomy, n (%)	43 (98)	1 (2.3)	44 (50)	<0.001
Postoperative data				
Median postoperative hospital stay, nights (range)	11 (7–30)	10 (6–33)	10 (6–33)	0.207
ICU for at least one night, n (%)	13 (30)	3 (6.8)	16 (18)	0.006
Median ICU stay, nights (range)	0 (0–26)	0 (0–2)	0 (0–26)	0.006
Postoperative transfusions, n (%)	22 (50)	10 (23)	32 (36)	0.008
Median postoperative transfusion, units (range)	0.5 (0–15)	0 (0–5)	0 (0–15)	0.012
Catecholamine administration, n (%) ^3^	17 (39)	5 (11)	22 (25)	0.003
Median catecholamine administration, h (range)	0 (0–600)	0 (0–61)	0 (0–600)	0.004
Catecholamine administration > 24 h, n (%) ^2^	6 (14)	3 (6.8)	9 (10)	0.484
Postoperative ventilation, n (%)	14 (32)	3 (6.8)	17 (19)	0.006
Median postoperative ventilation, h (range)	0 (0–602)	0 (0–5.8)	0 (0–602)	0.003

ICU = intensive care unit; TT = tumor thrombus. ^1^ Overall n = 76, TT group n = 41, non-TT group n = 35; ^2^ all patients with TT in the atrium; ^3^ the administration of catecholamines for <24 h was defined as standard and not considered as a complication.

**Table 3 cancers-16-03523-t003:** Sites of recorded complications and the assigned CDC grades.

Site of Complication	CDC	Complications
bleeding	II	erythrocyte deficiency, albumin deficiency, iron deficiency, vitamin K deficiency, anemia, and hematoma in the renal lodge
	IIIb	hematoma in the renal lodge
cardiovascular	I	hypotension, syncope, shock, arrhythmia, and leg ulcer
	II	hypertension, hypotension, circulatory depression with catecholamine requirements, shock, arrhythmia, leg ulcer, pericardial effusion, and thrombosis
	IIIb	myocardial infarction and pericardial effusion
	IVa	circulatory failure (resuscitation) and tachyarrhythmia absoluta (cardioversion)
gastrointestinal	I	nausea, vomiting, and diarrhea
	II	diarrhea, intestinal atony/ileus, intestinal infection, and bloated abdomen
urinary tract	I	oliguria, diuresis reduced/concentrated, retention parameters increased, renal insufficiency, and urinary tract infection
	II	renal insufficiency and urinary tract infection
	IVa	renal failure
pulmonary	I	pneumonia, pleural effusion, bronchitis, dyspnea, and cough
	II	pneumonia, pulmonary embolism, pleural effusion, bronchitis, dyspnea, cough, and prolonged weaning
	IIIa	pleural effusion
	IIIb	tube failure
	IVa	pneumonia and respiratory failure (intubation)
wound	I	wound infection, wound healing disorder, and abscess
	II	wound infection, wound healing disorder, and abscess
	IIIa	abscess
	IIIb	abscess
neurological/psychiatric	I	sensory disturbances, somnolence, unconsciousness, and radial paresis
	II	delirium, restless legs syndrome, and anxiety/panic attacks
dysglycemia	II	hypoglycemia, hyperglycemia, and blood glucose derailments
electrolyte imbalances *	I	hyperkalemia, hypokalemia, hypercalcemia, hypocalcemia, and hyponatremia
	II	hyperkalemia
other	I	edema (hands, legs, whole body, and genital), decubitus, and ascites
	II	edema (hands, legs, whole body, and genital), allergy, hyperuricemia, ascites, peritonitis, bacteremia, fever of unknown origin, and sepsis
	IIIa	ascites
	IVa	sepsis
infections (reported/graded within the groups above)	pneumonia, bronchitis, urinary tract infection, intestinal infection, wound infection, wound healing disorder, abscess, bacteremia, fever of unknown origin, and sepsis	

CDC = Clavien–Dindo classification. * In similar publications, electrolyte imbalances with the substitution of electrolytes were not listed as complications according to the CDC. To ensure comparability, we have not included these here either but present them in Appendix A.

**Table 4 cancers-16-03523-t004:** Complications and CDC grades.

	TT Group	Non-TT Group	Overall	*p*-Value
	(n = 44)	(n = 44)	(n = 88)	(TT vs. Non-TT)
Complications				
Patients with ≥1 complication (any), n (%)	41 (93)	32 (73)	73 (83)	0.021
≥1 bleeding complication, n (%)	29 (66)	14 (32)	43 (49)	0.001
≥1 cardiovascular complication, n (%)	25 (57)	13 (30)	38 (43)	0.010
≥1 gastrointestinal complication, n (%)	22 (50)	10 (23)	32 (36)	0.008
≥1 urinary tract complication, n (%)	12 (27)	13 (30)	25 (28)	0.813
≥1 pulmonary complication, n (%)	13 (30)	8 (18)	21 (24)	0.211
≥1 infection, n (%)	10 (23)	8 (18)	18 (20)	0.597
≥1 wound complication, n (%)	1 (2.3)	9 (20)	10 (11)	0.015
≥1 neurological/psychiatric complication, n (%)	8 (18)	1 (2.3)	9 (10)	0.030
≥1 dysglycemia, n (%)	5 (11)	0 (0)	5 (5.7)	0.055
≥1 other complication, n (%)	11 (25)	3 (6.8)	14 (16)	0.039
Median overall complications per patient (range)	3 (0–20)	1 (0–16)	2 (0–20)	< 0.001
CDC grades				
Grade 1				
antiemetics, n (%)	17 (39)	9 (20)	26 (30)	0.062
diuretics, n (%)	11 (25)	9 (20)	20 (23)	0.611
specific physiotherapy, n (%)	13 (30)	7 (16)	20 (23)	0.127
additional wound treatment, n (%)	1 (2.3)	9 (20)	10 (11)	0.015
antipyretics, n (%)	4 (9.1)	1 (2.3)	5 (5.7)	0.360
Grade 2				
medication other than for grade 1 complications, n (%)	35 (80)	22 (50)	57 (65)	0.004
blood products, n (%)	23 (52)	10 (23)	33 (38)	0.004
Grade 3				
surgical intervention (grade 3a), n (%)	1 (2.3)	1 (2.3)	2 (2.3)	1.000
surgical intervention (grade 3b), n (%)	1 (2.3)	1 (2.3)	2 (2.3)	1.000
endoscopic intervention (grade 3b), n (%)	1 (2.3)	1 (2.3)	2 (2.3)	1.000
Grade 4				
single organ dysfunction (grade 4a), n (%)	4 (9.1)	2 (4.5)	6 (6.8)	0.676
Multi-organ dysfunction (grade 4b), n (%)	1 (2.3)	0 (0)	1 (1.1)	1.000
Highest CDC grades				
No complications, n (%)	3 (6.8)	12 (27)	15 (17)	0.008
Highest CDC grade 1, n (%)	3 (6.8)	8 (18)	11 (13)	
Highest CDC grade 2, n (%)	33 (75)	20 (45)	53 (60)	
Highest CDC grade 3a, n (%)	1 (2.3)	0 (0)	1 (1.1)	
Highest CDC grade 3b, n (%)	0 (0)	2 (4.5)	2 (2.3)	
Highest CDC grade 4a, n (%)	3 (6.8)	2 (4.5)	5 (5.7)	
Highest CDC grade 4b, n (%)	1 (2.3)	0 (0)	1 (1.1)	
Major complications (CDC Grade ≥ 3), n (%)	5 (11)	4 (9.1)	9 (10)	1.000

CDC = Clavien–Dindo classification; TT = tumor thrombus.

## Data Availability

The data presented in this study are available on request from the corresponding author.

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
