# Peer review of "Complications of Tumor Nephrectomy with and Without Tumor Thrombus in the Vena Cava, Recorded with the Clavien–Dindo Classification: A Matched-Pair Analysis"

_cancers, 2024, doi:10.3390/cancers16203523_

Round 1

Reviewer 1 Report

Comments and Suggestions for Authors

The study analyzes the complications related to nephrectomies for renal cell carcinoma in patients with and without tumor thrombus, recorded with the Clavien-Dindo classification.

Introduction
The introduction is too brief and should be expanded to specifically discuss the tumor thrombus of renal cell carcinoma.

Paragraph 2.3 - Study design and data collection
The inclusion and exclusion criteria are unclear. Explain these criteria in more detail, especially the reason for excluding patients with tumors smaller than 3.8 cm.

Paragraph 2.3 - Surgical procedures
Specify the number of urologists involved and each one's years of experience.

Paragraph 2.6 - Statistical analyses
Explain the propensity score matching in a precise manner.

The authors should specify the histological subtype of renal cell carcinoma

Reviewer 2 Report

Comments and Suggestions for Authors

This study provides us with a comprehensive insight of the complications that arise following nephrectomies for RCC with and without a tumor thrombus. This study is significant as the authors have used a new method to classify these complications this will help clinicians to decide treatment options for future patients. While the paper is valuable to the field , I found the following minor weaknesses -

1. How is the CDC classification system better than the previously used one - the authors fail to highlight that.

2. Did they see any difference in complications between the male and female patients ?

3. There are grammatical errors and spell errors like - Were categorized complications to cardiovascular should be we categorized ?

4. The chinese words after some sentences should either be removed or translation should be added for the final manuscript.

Comments on the Quality of English Language

There are grammatical errors and spell errors like - Were categorized complications to cardiovascular should be we categorized ?

The chinese words after some sentences should either be removed or translation should be added for the final manuscript.

Reviewer 3 Report

Comments and Suggestions for Authors

The study aimed to analyze and compare postoperative complications in patients with renal cell carcinoma undergoing radical nephrectomy (RN) with and without inferior vena cava throm-bectomy (IVCT). It found that significantly more patients in the TT group experienced postoperative complications compared to the non-TT group, with higher incidences of cardiovascular, gastrointestinal, neurological/psychiatric, and bleeding complications. The study's limitations included a small sample size and its retrospective nature, emphasizing the need for experienced surgeons and trained staff to ensure early recognition and treatment of complications.

Although this study was conducted at a single institution (Jena University Hospital) with a retrospective design, I believe it is a valuable study that covered even minor postoperative complications.

Major points

Although propensity score matching has been used, important differences in baseline characteristics such as tumor size and the presence or absence of metastasis remained. Therefore I recommend to add this limitation in Discussion section.

Minor points

Please remove Chinese sentences (page 2, 4-6).
